# Physical Activity, Cardiorespiratory Fitness, and the Metabolic Syndrome

**DOI:** 10.3390/nu11071652

**Published:** 2019-07-19

**Authors:** Jonathan Myers, Peter Kokkinos, Eric Nyelin

**Affiliations:** 1Cardiology Division, Veterans Affairs Palo Alto Health Care System and Stanford University, Stanford, CA 94304, USA; 2Cardiology Division, Washington DC Veterans Affairs Medical Center and Rutgers University, Washington, DC 20422, USA; 3Endocrinology Division, Washington DC Veterans Affairs Medical Center, Washington, DC 20422, USA

**Keywords:** metabolic syndrome, cardiorespiratory fitness, insulin resistance, cardiovascular disease, exercise training

## Abstract

Both observational and interventional studies suggest an important role for physical activity and higher fitness in mitigating the metabolic syndrome. Each component of the metabolic syndrome is, to a certain extent, favorably influenced by interventions that include physical activity. Given that the prevalence of the metabolic syndrome and its individual components (particularly obesity and insulin resistance) has increased significantly in recent decades, guidelines from various professional organizations have called for greater efforts to reduce the incidence of this condition and its components. While physical activity interventions that lead to improved fitness cannot be expected to normalize insulin resistance, lipid disorders, or obesity, the combined effect of increasing activity on these risk markers, an improvement in fitness, or both, has been shown to have a major impact on health outcomes related to the metabolic syndrome. Exercise therapy is a cost-effective intervention to both prevent and mitigate the impact of the metabolic syndrome, but it remains underutilized. In the current article, an overview of the effects of physical activity and higher fitness on the metabolic syndrome is provided, along with a discussion of the mechanisms underlying the benefits of being more fit or more physically active in the prevention and treatment of the metabolic syndrome.

## 1. Overview

Chronic, non-communicable diseases currently represent the predominant challenge to global health. In a recent global status report on chronic disease, the World Health Organization stated that non-communicable conditions, including cardiovascular disease (CVD), diabetes and obesity, now account for roughly two-thirds of deaths worldwide [1]. The prevalence of many of the components of the “metabolic syndrome”, particularly obesity and diabetes, has grown considerably throughout the Western World since this term was initially suggested by Haller in 1977 [2]. Given that the metabolic syndrome is an important precursor to CVD and other chronic conditions [3,4,5,6,7], guidelines from various professional organizations have called for greater efforts to reduce the incidence of this condition and its components [3,4]. A notable parallel over the last 4 decades is the fact that numerous surveys and cohort studies have consistently reported that Western societies are significantly less physically active than past generations [7,8,9,10,11]. Moreover, a growing number of studies has reported that higher cardiorespiratory fitness (CRF; which is defined as the maximal capacity of the cardiovascular and respiratory systems to supply oxygen to the skeletal muscles during exercise) is inversely related to the development of the metabolic syndrome [12,13]. These studies, along with recent intervention trials [14,15], suggest a compelling link between impaired CRF, low physical activity patterns (defined as movement that requires energy), exercise (defined as planned, structured, repetitive, intentional movement intended to improve CRF), and the metabolic syndrome. 

Although the metabolic syndrome is complex and has been defined differently by different organizations, the clustering of risk factors that define it (high waist circumference, dyslipidemia, hypertension, and insulin resistance) are, to a certain extent, commonly associated with sedentary lifestyles. Indeed, numerous studies in recent decades have shown that increasing amounts of physical activity and higher CRF have a favorable impact on each of the components of the metabolic syndrome [12,13,14,15,16]. While physical activity interventions alone cannot be expected to normalize insulin resistance, lipid disorders, or obesity, the combined effect of increasing activity on these risk markers, an improvement in CRF, or both, can have a major impact on health outcomes related to the metabolic syndrome. However, physical activity as a treatment for metabolic disease remains underutilized. In fact, physical activity interventions are often dismissed in favor of pharmacologic treatments or other interventions that tend to be more economically driven [8,17,18,19]. Although physical activity counseling is now mandated by many health care systems, the fact remains that activity counseling rarely occurs as part of clinical encounters [20,21]. The lack of attention paid to physical activity is unfortunate given the strength of exercise interventions on health outcomes among individuals with metabolic disorders [12,13,14,15,16]. 

In the following, an overview of the effects of physical activity and CRF on the metabolic syndrome is provided, along with a discussion of the mechanisms underlying the benefits of being more fit or more physically active in the prevention and treatment of the metabolic syndrome. 

## 2. Physical Activity and the Metabolic Syndrome 

Collectively, studies on the impact of being more physically active, whether studied in a cross-sectional cohort or as a result of a structured exercise intervention, have been shown to have an important impact on cardiometabolic risk. Regular exercise can help to reduce weight, reduce blood pressure, and improve lipid disorders, including raising HDL and lowering triglycerides [7,16,17,18,19,20,21,22]. Among the physiological systems that respond favorably to physical activity, it has been argued that one of the most demonstrable effects of regular exercise is its impact on insulin resistance [23,24]. A summary of key studies is shown in Table 1; notably, these studies are categorically consistent in demonstrating the benefits of being more physically active in terms of reducing risk for the metabolic syndrome. 

### 2.1. Observational Studies Associating Physical Activity Patterns with Metabolic Risk

Observational or cross-sectional studies are inherently limited because they do not demonstrate cause and effect. In the current context, the weaknesses of these studies include the fact that intrinsically healthier individuals may be more likely to engage in physical activity, or that they may be genetically more fit irrespective of lifestyle or behavioral factors. Nevertheless, these studies have provided valuable information regarding patterns between physical activity habits, metabolic risk, and related conditions. Collectively, these studies suggest that more active individuals exhibit either a lower prevalence of risk factors for the metabolic syndrome, have a lower incidence of developing the metabolic syndrome over a given follow-up period, or both. While the levels of activity have been quantified and defined in different ways, these data support the concept that meeting the minimal guidelines on activity (i.e., 150 minutes per week of moderate intensity activity) is associated with a lower prevalence of the metabolic syndrome. In the following, a sampling of some of the key observational studies related to physical activity and the metabolic syndrome are outlined.

As part of the TROMSO study in Norway, Thune and colleagues [25] studied 5220 men and 5869 women who completed two physical activity surveys approximately 7 years apart. BMI and detailed lipid profiles were determined at both evaluations. There was a dose–response relationship between improved serum lipid levels, BMI, and higher levels of physical activity in both genders after adjustments for potential confounders. Differences in BMI and serum lipid levels between sedentary and sustained exercising groups were consistently more pronounced after 7 years than at baseline, especially in the oldest age group. The most dramatic differences in metabolic risk profiles occurred between the most active subjects compared to the least active subjects. An increase in leisure time activity over the 7 years improved metabolic profiles, whereas a decrease worsened them in both genders.

In a cross-sectional evaluation of physical activity and metabolic risk among individuals with a family history of Type 2 diabetes, Ekelund et al. [29] measured total body movement and five other subcomponents of physical activity by accelerometry in 258 at-risk adults. Body composition was determined using bioimpedance and waist circumference, and blood pressure, fasting triglycerides, HDL, glucose, and insulin were determined. In addition, continuously distributed clustered risk was calculated. Total body movement (counts/day) was significantly and independently associated with three of six risk factors (fasting triglycerides, insulin, and HDL) and with clustered metabolic risk after adjustment for age, gender, and obesity. Time spent at moderate- and vigorous-intensity physical activity was independently associated with clustered metabolic risk. Short (5- and 10-minute) bouts of activity, time spent sedentary, and time spent at light-intensity activity were not significantly related with clustered risk after adjustment for confounding factors. The association between total body movement and intermediary phenotypic risk factors for cardiovascular and metabolic disease along with clustered metabolic risk was independent of aerobic fitness and obesity. These investigators suggested that increasing the total amount of physical activity in sedentary and overweight individuals has beneficial effects on metabolic risk.

Laaksonen and colleagues [26] assessed 12-month leisure time physical activity (LTPA), VO_2_max, and cardiovascular and metabolic risk factors among 612 middle-aged men without the metabolic syndrome at baseline. After 4 years of follow-up, 107 men had metabolic syndrome (using the WHO definition). Men engaging in 3 h/week of moderate or vigorous LTPA were half as likely as sedentary men to have the metabolic syndrome after adjustment for major confounders (age, BMI, smoking, alcohol, and socioeconomic status) or potentially mediating factors (insulin, glucose, lipids, and blood pressure). Vigorous LTPA had an even stronger inverse association with incidence of the metabolic syndrome among men who were unfit at baseline. Men in the upper tertile of VO_2_max were 75% less likely than unfit men to develop the metabolic syndrome, even after adjustment for major confounders. Associations of LTPA and VO_2_max with development of the metabolic syndrome were qualitatively similar. These results suggest that high-risk men engaging in commonly recommended levels of physical activity were less likely to develop the metabolic syndrome than sedentary men. CRF was also strongly protective, although possibly not independent of mediating factors.

### 2.2. Exercise Intervention Studies and the Metabolic Syndrome

Relative to cross-sectional studies, exercise and lifestyle intervention studies can provide more direct information on the cause and effect impact of physical activity, CRF, or both, on risk of the metabolic syndrome. While there is a lengthy history of studies applying exercise interventions to assess the effects of training on individual components of the metabolic syndrome (e.g., insulin resistance, blood pressure, abdominal adiposity), fewer studies have been specifically designed to examine the efficacy of exercise training on the clinical diagnosis or reversal of the metabolic syndrome. In recent years, a growing number of groups have conducted large, multicenter randomized trials of exercise training along with other lifestyle interventions among individuals with or at high risk for the metabolic syndrome. 

Two large lifestyle intervention trials, the Finnish Diabetes Prevention Study (DPS) [35], and the US Diabetes Prevention Program (DPP) [36], were designed to either prevent type 2 diabetes in impaired glucose-tolerant subjects or reduce the prevalence of metabolic syndrome through changes in diet and physical activity. The DPS observed a 58% reduction in risk for the development of type 2 diabetes with lifestyle intervention, and the DPP demonstrated a reduced prevalence of the metabolic syndrome in the intervention group. Weight loss appeared to be a major determinant of both improvements in glucose tolerance and the reduction in metabolic syndrome prevalence, whereas physical activity and dietary composition contributed independently. A third trial performed in the Netherlands involved a lifestyle intervention designed to assess the impact of diet and physical activity intervention on glucose tolerance in impaired glucose-tolerant subjects (termed the Study of Lifestyle intervention and Impaired glucose tolerance Maastricht [SLIM] study) [37]. The SLIM study similarly reported a 58% reduction in diabetes risk after 3 years and a 47% reduction at the end of the intervention, despite a relatively modest weight reduction. A follow-up to the SLIM trial determined the effects of the exercise and lifestyle intervention on the incidence and prevalence of the metabolic syndrome during the active intervention and four years thereafter [38]. They observed that the prevalence of the metabolic syndrome was significantly lower in the intervention group (52.6%) compared to the control group (74.6%). In addition, among participants without the metabolic syndrome at baseline, cumulative incidence of the metabolic syndrome was 18.2% in the intervention group at the end of active intervention, compared to 73.7% in the control group. Four years after stopping active intervention, the reduced incidence of metabolic syndrome was maintained.

A landmark multicenter trial performed in the US, termed Action for Health in Diabetes (Look AHEAD) [30], assessed whether an intensive lifestyle intervention for weight loss would decrease cardiovascular morbidity and mortality in patients with Type 2 diabetes. In 16 study centers in the US, 5145 overweight or obese patients with type 2 diabetes were randomly assigned to participate in an intensive lifestyle intervention that promoted weight loss through decreased caloric intake and increased physical activity (intervention group) or to receive diabetes support and education (control group). The primary outcome was a composite of death from cardiovascular causes, nonfatal myocardial infarction, nonfatal stroke, or hospitalization for angina during a maximum follow-up of 13.5 years. The trial was stopped early on the basis of a futility analysis at a median follow-up of 9.6 years. Weight loss was greater in the intervention group than in the control group throughout the study (8.6% vs. 0.7% at 1 year; 6.0% vs. 3.5% at study end). The intensive lifestyle intervention also produced greater reductions in HbA1c and greater initial improvements in fitness and all cardiovascular risk factors, except for LDL cholesterol. The primary outcome occurred in 403 patients in the intervention group and in 418 in the control group (1.83 and 1.92 events per 100 person-years, respectively); these differences were not significant (*p* = 0.51). 

While the Look AHEAD study did not reduce the rate of cardiovascular events in overweight or obese adults with type 2 diabetes, there were many notable benefits among subjects in the intervention group. These included the fact that modest weight loss occurred and was maintained over 10 years, clinically meaningful improvements in HbA1c which were greatest during the first year but were at least partly sustained throughout follow-up, fewer subjects needing treatment with insulin, partial remission of diabetes during the first 4 years of the trial vs. control subjects, reduced sleep apnea and depression, and improvements in quality of life, physical functioning, and mobility.

There are also numerous notable single-center trials that have assessed the impact of exercise intervention on metabolic risk. Stewart et al. [31] studied 51 men and 53 women with or at elevated risk for metabolic syndrome who underwent either a 6-month supervised exercise program or usual care. Exercise significantly increased aerobic and muscle fitness, lean mass, and HDL, and reduced total and abdominal fat. Reductions in total body and abdominal fat and increases in leanness, largely independent of weight loss, were associated with improved systolic and diastolic blood pressure, total cholesterol, very low-density lipoprotein cholesterol, triglycerides, lipoprotein(a), and insulin sensitivity. At baseline, 42.3% of participants had metabolic syndrome. At 6 months, nine exercisers (17.7%) and eight controls (15.1%) no longer had metabolic syndrome, whereas four controls (7.6%) and no exercisers developed it.

Katzmarzyk, et al. [32] studied the efficacy of exercise training in treating the metabolic syndrome among 621 participants from the HERITAGE Family Study, identified at baseline as sedentary but apparently healthy. Subjects underwent a 20-week program of exercise training consisting of 3 sessions/week of supervised cycle ergometer training. The presence of the metabolic syndrome and the cluster of associated risk factors were determined before and after the study period. Exercise training resulted in marked improvements in the metabolic profile of the participants, including triglycerides, HDL cholesterol, blood pressure, fasting plasma glucose, and waist circumference. Of the 105 participants with the metabolic syndrome at baseline, 30.5% (32 participants) were no longer classified as having the metabolic syndrome after training. There were no sex or race differences in the efficacy of exercise in treating the metabolic syndrome.

### 2.3. Meta-Analyses of Exercise and Cardiometabolic Risk

There have been many individual trials in the context of physical activity and the metabolic syndrome, and many have lacked adequate sample sizes. The metabolic syndrome is more complex than many other conditions because it involves the clustering of several risk factors, and has been defined in different ways. Some studies have reported a significant effect on one or several risk factors but a minimal effect on another. Meta-analyses have been particularly helpful in this area by combining results from different studies to obtain a better estimate of the overall effect of a particular intervention. There have been several notable meta-analyses in the area of physical activity and metabolic syndrome which are discussed in the following. 

Wewege et al. [39] recently performed a meta-analysis examining the effect of aerobic, resistance and combined (aerobic and resistance) exercise on cardiovascular risk factors among individuals with the metabolic syndrome, but without a diagnosis of diabetes. Interestingly, this is an understudied group, yet it represents the majority of the metabolic syndrome population. Randomized controlled trials >4 weeks in duration that compared an exercise intervention to non-exercise control groups in patients with metabolic syndrome without diabetes were included. Eleven studies with 16 interventions were analyzed (12 aerobic, 4 resistance). Aerobic exercise significantly improved waist circumference, fasting glucose, HDL cholesterol, triglycerides, diastolic blood pressure, and cardiorespiratory fitness (by 4.2 mL/kg/min, *p* < 0.01), among other outcomes. No significant effects were determined following resistance exercise possibly due to limited data. Sub-analyses suggested that aerobic exercise that progressed to vigorous intensity, and conducted 3 days/week for ≥12 weeks, offered larger and more widespread improvements. While these results strongly support the use of aerobic exercise for patients with the metabolic syndrome who have not yet developed diabetes, they also suggest that more studies on resistance/combined exercise programs are required to improve the quality of evidence.

Naci and Ioannidis [40] performed a recent meta-analysis among 14,716 subjects randomized to either a physical activity intervention or usual care. While the analysis did not address metabolic syndrome per se, the results are remarkable in that they provided a direct comparison between exercise and drug therapies for diabetes and CVD risk. Among 57 trials comparing the effects of drug and physical activity interventions on health outcomes compared to usual care, they observed that exercise intervention was similar to drug interventions for the secondary prevention of prediabetes, cardiovascular disease (CVD) and mortality. Importantly, physical activity was markedly superior to drug treatment among patients with stroke. The extent to which standard pharmacologic treatment would complement exercise interventions in treating or preventing diabetes or other health outcomes is unknown since there are so few data on comparative effectiveness involving exercise. These results are striking given the investments made in drug interventions relative to the comparatively meager investments devoted to exercise and other preventive strategies. 

Ostman and colleagues [41] performed a meta-analysis that included 16 studies with 23 intervention groups and a total of 77,000 patient-hours of exercise training. All studies included subjects with a clinical diagnosis of the metabolic syndrome at baseline, an intervention involving exercise vs. sedentary controls, and all studies included incidence of mortality and hospitalization. Exercise training duration ranged between 8 weeks and 1 year. In analyses comparing aerobic exercise training versus control groups, there were reductions in BMI, waist circumference, systolic blood pressure and diastolic blood pressure, fasting blood glucose, triglycerides and low-density lipoprotein. Peak VO_2_ was significantly improved among those randomized to exercise (mean difference 3.0 mL/kg/min, *p* < 0.001). Similar changes were observed for studies using combined aerobic and resistance exercise. 

### 2.4. Synopsis—Physical Activity and the Metabolic Syndrome 

Higher levels of physical activity, whether through observational studies or as part of formal exercise intervention trials, generally have a favorable impact on the metabolic syndrome and its components. In some studies, the proportion of participants who meet the criteria for the metabolic syndrome is reduced with exercise intervention. In longitudinal studies, more active individuals have a lower incidence of the metabolic syndrome. In a limited number of studies in which the dose–response relationship has been assessed, the most active subjects tend to have the greatest reductions in metabolic risk. Although the dose of physical activity has varied in the different studies, achieving the minimal physical activity guidelines (at least 150 minutes per week of moderate-intensity activity or 75 minutes per week of vigorous intensity activity) has been consistently demonstrated to have significant benefits on metabolic risk. While there are comparatively few studies on the impact of strength training on cardiometabolic risk, higher levels of muscular strength are associated with lower risk for developing the metabolic syndrome. Thus, in addition to aerobic exercise, individuals should strive to achieve the minimal recommendations of at least 2 days per week of resistance training.

## 3. Cardiorespiratory Fitness and the Metabolic Syndrome

Some of the inconsistencies between metabolic syndrome incidence and self-reported physical activity status [26,42] may be explained by the subjectivity and inaccuracy of self-reported physical activity assessments [43]. In this regard, directly measured or estimated VO_2_ max based on a standardized exercise treadmill or cycle ergometer represents an objective assessment of CRF, as subject bias in reporting physical activity is removed. Overall, such studies have been consistent in reporting a lower prevalence of metabolic syndrome in those with higher CRF among both men and women regardless of race and after adjustment for relevant confounders [44,45,46,47,48]. An overview of some of the key studies is provided in Table 2.

In Finland, men and women in the lowest tertile of VO_2_max had 10.2 and 10.8 times, respectively, higher risks of having metabolic syndrome than those in the highest VO_2_max category [44]. Similar findings were reported by Kelley et al. [45] in middle-aged men and women in the US. Importantly, an inverse and graded association between CRF and the incidence of metabolic syndrome has been observed with relatively small changes in CRF (e.g., 2.5 metabolic equivalents) yielding significant reductions in risk. The risk of metabolic syndrome prevalence was more than 20 times less likely for individuals in the highest fit category compared to the least-fit individuals. Similarly, the risk of developing metabolic syndrome within 2 years of follow-up was reported to be 44% lower for each 1-SD increase in VO_2_ max [48]. Individuals in the highest sex-specific fitness category were 68% less likely to develop metabolic syndrome. In a more recent Finnish study, each 1-SD higher change in VO_2_ max from baseline was associated with a 1.8-fold higher likelihood of resolving metabolic syndrome during 2 years of follow-up [48].

An interaction between CRF risk of and metabolic syndrome has also been suggested by the Australian National Health Survey. An inverse association between CRF and the metabolic syndrome was observed in those with lower waist circumference and fasting glucose in both men and women [51]. However, the impact of CRF on metabolic syndrome was independent of obesity as defined by body mass index among 9666 middle-aged (48.7 ± 8.4 years) asymptomatic men [52]. The likelihood of developing metabolic syndrome was approximately 50% lower in fit men compared to unfit men (OR = 0.51, 95% CI 0.46 to 0.57), independent of BMI, particularly in men <50 years. There is also evidence to suggest that objectively measured energy expenditure is a stronger deterrent for the metabolic syndrome than measured VO_2_ max. Franks and colleagues [50] reported that the magnitude of the association between physical activity and the metabolic syndrome was >3-fold greater than for VO_2_max, suggesting that the risk of metabolic syndrome can be modulated by higher intensity activities as well as lower intensity aerobic activities (below the threshold required to increase aerobic capacity). 

Some evidence also suggests that the CRF-metabolic syndrome risk association may be gender-specific. Hassinen et al. [48] observed that each 1-SD higher VO_2_ max (6.1 mL/kg/min in men; 4.8 mL/kg/min in women) resulted in 56% and 35% decreased risks of developing metabolic syndrome over two years of follow-up in men and women, respectively. In men, a 4.8 mL/kg/min increase in VO_2_ max resulted in a 56% decrease in risk [48]. However, gender differences in the association between CRF and incidence of metabolic syndrome were not supported by the same group in their previous study [44]. 

There are also indications that only high CRF may offer protection against the development of metabolic syndrome. Laaksonen et al. [26] reported 47% and 75% lower odds of developing metabolic syndrome among men in the middle and highest tertiles of measured VO_2_max, respectively, compared to men in the lowest tertile. However, this association was no longer significant after adjustment for baseline metabolic risk factors. Similarly, Carnethon et al. [49] reported that only men and women in the highest 40% of maximal treadmill performance were protected against developing metabolic syndrome. In contrast, LaMonte et al. [46] reported an independent and progressive decline in the risk of developing metabolic syndrome with increased CRF for men and women. Specifically, they reported 20% to 26% lower risks among participants with moderate CRF levels and 53% to 63% lower risks in the highest CRF categories, when compared to those in the lowest CRF category. 

Finally, in the Diabetes Prevention Program trial, 3234 subjects (53% with metabolic syndrome) at high risk for diabetes were randomized to a lifestyle intervention group or usual care [34]. Those in the lifestyle intervention group (aerobic exercise 150 minutes per week and nutritional counseling) achieved a 38% reversal of the metabolic syndrome and a 41% reduction of new onset of metabolic syndrome. In contrast, treatment with metformin only reduced new cases of metabolic syndrome by 17%. To prevent one case of diabetes during a period of three years, 6.9 persons would have had to participate in the lifestyle-intervention program, and 13.9 would have had to receive metformin. These findings suggest that a healthy lifestyle may be more effective in preventing metabolic syndrome than the anti-hyperglycemic agent metformin.

### Synopsis—Cardiorespiratory Fitness and the Metabolic Syndrome 

Although physical activity and CRF are often used interchangeably, it is important to recognize that they are different; physical activity is a behavior and CRF is an attribute. CRF is improved by activity, but it is influenced by other factors, including genetics. Nevertheless, most sedentary individuals will improve CRF by following the widely-recognized minimal guidelines on physical activity. Both CRF levels from observational studies and changes in CRF as a result of 3–12-month exercise interventions have consistently been shown to improve cardiometabolic risk. In some studies, a proportion of a study sample no longer meets the criteria for metabolic syndrome after an exercise intervention that increases CRF. Taken together, cross-sectional studies demonstrate that subjects in the highest-fit categories exhibit between 5- and 20-fold lower likelihood of having the metabolic syndrome vs. subjects in the least-fit groups. Longitudinally, subjects in the highest fit groups exhibit ≈40% to as much as 20-fold lower risks of developing the metabolic syndrome. Efforts to improve CRF should be part of standard therapy for individuals with or at high risk for the metabolic syndrome. 

## 4. Mechanisms Underlying the Metabolic Syndrome and Implications for Physical Activity and Fitness 

### 4.1. Pathophysiology of Metabolic Syndrome

The underlying cause(s) of the metabolic syndrome are unknown, but it is significantly influenced by the twin epidemics of diabetes and obesity. Indeed, metabolic syndrome shares diabetes-related insulin resistance (IR) and dysfunctional adipose fuel handling and central obesity as its core. However, despite the commonality of traits, many subjects with metabolic syndrome do not display IR [53] nor do all obese individuals have metabolic syndrome [54,55]. Thus, neither IR nor central obesity fully explains the pathophysiological features of metabolic syndrome and other factors implicated include inflammation, genetics, epigenetics, and circadian abnormalities. As will be discussed further, enhanced CRF modulates the negative impact of these causative drivers of metabolic syndrome.

### 4.2. Insulin Resistance

The concept of IR was introduced by Himmsworth in 1936 who showed that diabetes could be subdivided into two categories—insulin-sensitive and insulin-insensitive types [56]—and this was later confirmed by Yalow and Berson with the novel measurement of insulin itself [57]. Once clamp techniques were developed, it was established that IR predominated in type 2 diabetes [58,59] and that hyperinsulinemia was the best predictor of the development of type 2 diabetes in nondiabetic individuals [60]. Reaven coined the term Syndrome X, later renamed by others to metabolic syndrome, to describe the role of IR (i.e., hyperinsulinemia or impaired glucose tolerance) as the driver of atherosclerotic dyslipidemia, type 2 diabetes, and hypertension [61]. Indeed, elevations of insulin concentration were shown to prospectively precede the development of these metabolic disorders [62] and the role of IR in metabolic syndrome was shown to be related to low insulin sensitivity [63]. With an increasing degree of metabolic syndrome components (i.e., metabolic syndrome score), there is an increase in fasting glucose, insulin levels, and HOMA IR [64]. 

During physiological conditions, insulin binds to its receptor leading to tyrosine phosphorylation of downstream substrates including activation of the phosphoinositide 3-kinase (PI3K) pathway resulting in recruitment of GLUT4 to mediate glucose transport into muscle and adipose tissue where it is phosphorylated and either stored as glycogen or metabolized to produce ATP. However, when a state of compensatory hyperinsulinemia occurs in IR subjects, due to changes in insulin secretion and/or insulin clearance [65], the ensuing response includes mild forms of glucose intolerance, dyslipidemia (high triglycerides, low HDL, small dense LDL), and hypertension which is the pathophysiological construct of the insulin resistance syndrome developed by Reaven leading to increased risk of CVD, as well conditions such as stroke, polycystic ovary syndrome, non-alcoholic fatty liver disease, cancer, and sleep apnea [66]. Importantly, according to Reaven, the individual components of the IR syndrome can occur without IR, and the presence of IR does not have to lead to any of the components of the syndrome. Interestingly, although IR has been considered the central driver of type 2 diabetes and metabolic syndrome, an alternative argument places IR as an adaptive biomarker of poor metabolic health and insulin hyperesponsivness as the root cause [67]. 

The reason why IR leads to atherogenesis has been attributed to the activation by insulin of the mitogen activated protein (MAP) kinase pathway which, as opposed to the muted PI3K pathway, functions normally in IR. Subnormal PI3K-Akt activity leads to a reduction in endothelial nitric oxide formation and endothelial dysfunction, reduction in GLUT4 translocation, and decreased skeletal muscle and fat glucose uptake [68]. Concurrently, the persistence of MAP kinase activity results in augmented expression of endothelin-1 and endothelial adhesion molecules with vascular smooth muscle cell mitogenesis which leads to vascular abnormalities and increased atherosclerosis risk. 

The skeletal muscle mass comprises approximately 40% of total body mass and is the primary source of insulin-mediated glucose uptake and fatty acid oxidation. The exposure to exercise evokes adaptation in skeletal muscle in a multitude of signaling pathways, the functional response to which is determined by training volume, mode of training, intensity and frequency. With persistent exercise exposure, there is mitochondrial biogenesis, fast-to-slow fiber-type transformation, changes in substrate metabolism, and angiogenesis. Moreover, a host of myokines are released from active muscles providing communication throughout the body. Enhanced fitness is associated with high levels of insulin sensitivity/insulin action. While glucose homeostasis at rest is insulin-sensitive, exercise with muscle contractions increases glucose uptake from the circulation that is not reliant on insulin. Indeed, GLUT-4 is responsive to both insulin and muscle contraction independently.

Whatever the role of IR, it is known that exercise augments insulin signaling independent of PI3K and when skeletal muscles are stimulated by contraction combined with insulin, glucose transport and GLUT4 translocation are enhanced. Thus, exercise provides a potent means to avert metabolic syndrome supported by the results of the Diabetes Prevention Program study [36], where exercise intervention decreased metabolic syndrome prevalence significantly compared to the control group throughout the intervention. Interestingly, although both aerobic and resistance training increase glucose transport and are often additive, these metabolic responses appear to be mediated by different mechanisms. In at least one study among nonobese young women, the investigators reported that insulin sensitivity increases in both aerobically-trained and resistance-trained women. However, when data were expressed per kg of free-fat mass (FFM) the improvement in glucose disposal persisted in endurance-trained women, whereas no significant change was noted in resistance-trained subjects or controls. This led to the conclusion that the increased glucose disposal associated with resistance exercise was the result of the increase in the quantity of lean body mass, without altering the intrinsic capacity of the muscle to respond to insulin. On the other hand, endurance training enhanced glucose disposal independent of changes in lean body mass or VO_2_max, suggestive of an intrinsic change in the ability of the muscle to metabolize glucose [69]. 

### 4.3. Adipose Fuel Metabolism

The metabolic consequences related to an unhealthy lifestyle were proposed in 1923 by Kylin consisting of a syndrome of hypertension, hyperglycemia, hyperuricemia, and obesity [70]. “Androgenic obesity” contributing to diabetes and CVD was proposed later by Vague in 1940 [71]. The upper-body or abdominal obesity type with hyperinsulinemia, in particular, was proposed as the primary factor leading to metabolic syndrome and CVD independent of overall obesity [72,73,74]. Being a multifunctional organ providing cross-talk between various systems, including the immune and the cardiovascular systems, this type of abdominal obesity, being the most common manifestation of metabolic syndrome, has been viewed as a cellular biomarker of dysfunctional adipose tissue [55] or adiposopathy [75]. 

Insulin is the major regulator of fuel metabolism in adipocytes and is adversely impacted by excess caloric intake and inactivity. Hyperinsulinemia is well documented in individuals with obesity with or without IR and is related to increases in insulin secretion and decreases in insulin clearance rate [76]. It has been known for some time that insulin insensitivity can cause a distinct biochemical syndrome with elevated free fatty acids [77] and in metabolic syndrome-prone subjects, relative tissue hypoinsulinemia results in release of excess free fatty acids, mainly from visceral depots, resulting in increased liver synthesis VLDL, elevated triglycerides, increased HDL clearance and small dense LDL. Increased free fatty acid release also causes lR in the liver resulting in increased gluconeogenesis and hyperglycemia. These metabolic events result in adipocyte fuel malfunction manifested as adipocyte hypertrophy and ectopic lipid deposition in vital organs such as the liver, pancreas, muscle and heart. In the pancreas, lipid excess can lead to lipotoxicity which can promote endoplasmic reticulum stress-mediated β-cell death [78].

Adipose tissue also harbors fat-derived mesenchymal stem cells which experimentally have the capacity to modify mRNA expression contributing to IR [79]. Moreover, abdominal fat and fat-derived mesenchymal stem cells are responsive to physical activity; both high-intensity aerobic and resistance training decrease visceral fat effectively [80] while the molecular expression of fat-derived mesenchymal stem cells is significantly altered with exercise preventing adipogenesis [81].

### 4.4. Inflammation

Systemic inflammation has been strongly linked to CVD via multiple immune system biomarkers, factors also associated with metabolic syndrome. Thus, metabolic syndrome is associated with pro-inflammatory cytokines such as TNF, IL-beta, and is characterized by chronic systemic low-grade inflammation manifested by elevated CRP [82]. Chronic inflammation links metabolic syndrome to IR [83,84] and to CVD via promotion of vascular dysfunction [85]. Adipocyte hypertrophy, abnormal local blood flow, hypoxia, altered adipokine expression, and local infiltration of immune cells all conspire to adiposopathy which is infiltrated by macrophages, with elevated TNF and IL-6. Moreover, these macrophage-associated adipocytes are undergoing necrosis. A recent finding showed that the use of the anti-inflammatory agent colchicine significantly improved obesity-associated inflammatory variables in metabolic syndrome and appeared to be safe [86].

The degree of cardiorespiratory fitness in metabolic syndrome subjects has been shown to have inverse associations with CRP, IL-6, and IL-18, partially explained by the degree of abdominal obesity [87]. Using IL-18 as a biomarker of inflammation, aerobic exercise reduced inflammation which was not observed with resistance exercise despite a similar degree of loss of fat mass in metabolic syndrome subjects [88]. 

### 4.5. Genetics/Epigenetics

Inheritance plays a role in metabolic syndrome and its influence may range from 10% to 30% being strongest between waist circumference and IR, which has also been documented in twin studies [89]. Techniques such as linkage analysis, candidate gene approach, and genome-wide association (GWAS) studies have been applied to detect gene variants for metabolic syndrome focusing on loci for individual components such as obesity, dyslipidemia, hypertension, and diabetes [90]. For example, eight single nucleotide polymorphisms (SNPs) were associated with the dyslipidemia in metabolic syndrome [91]. In other studies, GWAS associations have shown that transcription factor 7-like 2 (TCF7L2), which is part of the Wnt signaling pathway, mediates metabolic syndrome trait susceptibility towards developing diabetes and dyslipidemia [92]. Another example is the caveolin-1 gene (CAV1) variant associated with IR which is also associated with metabolic syndrome, especially in non-obese subjects [93]. 

The concept of epigenetics, originally accredited to Waddington, has evolved to specify how gene activation or silencing influence gene expression without changing the DNA sequence itself and the epigenome include DNA methylation, histone modification, and various RNA-mediated processes. The epigenetic expression can be altered during development, during varying nutritional conditions, and by physical activity. One of the epigenetic mechanisms involves DNA methylation which results in a methyl group being attached to a cytosine pyrimidine ring and thereby influence gene expression especially when located in promotor regions. A burgeoning area of inquiry involves DNA methylation which has been reported to be related to several components of the metabolic syndrome [94] including an inverse association between levels of methylation and worsening of the metabolic syndrome [95]. Studies assessing global DNA methylation and also assessing methylation at specific genes related to lipid metabolism appear to be related to causation of the metabolic syndrome [96]. Epigenetic methylation changes related to physical activity have been reported to occur in the regions regulating peroxisome proliferator-activated receptor-1α, the master regulator of exercise-muscle activity, and also impact the adipose tissue response [97,98,99]. 

### 4.6. Circadian Disruption and Metabolic Syndrome

Disrupted diurnal rhythms due to excessive light or shift work can have profound and disruptive whole-body metabolic effects impacting most hormones that are normally governed by circadian rhythmicity so it is not surprising that these perturbations can lead to metabolic syndrome conditions with IR and obesity. Clock genes are expressed in adipose tissue and correlate to metabolic syndrome parameters [100]. A shortened sleep duration less than 6 hours has been associated with increased risk of metabolic syndrome and CVD [101] and meta analyses support these associations between sleep deprivation and risk for metabolic syndrome [102,103]. Moreover, the relationship of shortened sleep and metabolic syndrome may be dose related [104]. However, a normal sleep pattern decreases the risk of metabolic syndrome, albeit prolonged sleep appears to be neutral in this regard [105]. Regular exercise can re-set clock genes and have a salutary impact on clock time which might be another way to inhibit the metabolic syndrome [106,107]. A complementary medical strategy may be the use of a sympatholytic dopamine D2 receptor agonist to combat the circadian disruption and improve metabolic syndrome [108].

## 5. Summary

Both single center trials and recent meta-analyses suggest that exercise training, higher CRF, or both, improve factors that underlie the metabolic syndrome. Among subjects who meet the criteria for the metabolic syndrome, health outcomes are significantly improved by aerobic or resistance training, or their combination. In some individuals, an exercise program has been demonstrated to improve risk markers to an extent that they no longer meet the criteria for the metabolic syndrome. There are numerous physiological, lifestyle, and genetic factors that account for these salutary effects of physical activity or formal exercise programs. These include the impact of exercise on insulin resistance, adipose fuel metabolism, inflammation, and epigenetic factors. Physical activity interventions clearly have a favorable impact on metabolic disease and the burden it places not only on individuals but also on health care systems. Incorporating physical activity as an integral part of treatment strategies for the metabolic syndrome would appear to go a long way toward reducing the adverse health impact of this condition. 

## Figures and Tables

**Table 1 nutrients-11-01652-t001:** Sampling of studies assessing the impact of physical activity patterns or exercise intervention on the metabolic syndrome.

**Observational Studies**			
**Author, Year; (Reference)**	***N* (Men/Women),** **Mean Age**	**Assessment**	**Key Results**
Thune, 1998; [25]	5220/5869 34.4 and 33.7 years, respectively	PA self-report	Higher PA associated with better lipid profile, overall metabolic risk profile over 7 years
Laaksonen, 2002; [26]	612 men 51.4 years	Assessment of LTPA over previous 12 months among high risk men; followed for 4 years	>3 h/week moderate to vigorous LTPA half as likely as sedentary men to have MetSyn Men in top 33% VO_2_max 75% less likely than unfit men to develop MetSyn over 4 years
Sisson, 2010; [27]	697/749 47.5 years	Accelerometry	MetS prevalence decreased as steps/day increased; odds of having MetSyn were 10% lower for each additional 1000 steps/day
Healy, 2008; [28]	67/102 53.4 years	Accelerometer evaluation of time spent in sedentary, light, moderate-to-vigorous, and mean activity intensity in participants with diabetes and obesity	Moderate-to-vigorous activity associated with lower triglycerides. Sedentary time, light-intensity time, and exercise intensity associated with waist circumference and clustered metabolic risk
Ekelund, 2007; [29]	103/155 40.8 years	Accelerometry, exercise test, biometric measures on adults with a family history of type 2 diabetes	Total body movement inversely associated with triglycerides, insulin, HDL and clustered metabolic risk; moderate-and vigorous-intensity PA inversely associated with clustered metabolic risk
**Exercise Intervention Studies**			
**Author, Year**	***N***	**Intervention**	**Key Results**
Look AHEAD, 2013; [30]	3063/2082 58.8 years	Subjects with type 2 diabetes randomly assigned to intensive lifestyle intervention or diabetes support and education	Intervention group had greater reductions in weight loss, glycated hemoglobin and greater initial improvements in exercise capacity and all cardiovascular risk factors (except LDL)
Stewart, 2004; [31]	53/62 63.6 years	6 months of exercise training in subjects with or at high risk for MetSyn	Exercise group improved peak VO_2_, muscle strength, and lean body mass; reductions in total and abdominal fat related to improved CVD risk
Katzmarzyk, 2003; [32]	288/333 31.6	20 weeks of supervised aerobic exercise training	Of 105 patients with MetSyn, 30.5% were no longer classified as having metabolic syndrome after exercise training
Balducci, 2008; [33]	329/234	Twice weekly aerobic & resistance training for 1 year	Exercise group improved fitness, HbA1c, and CVD risk profile
Diabetes Prevention Program Research Group, 2002; [34]	3234 50.6	Lifestyle intervention (150 min/week PA and nutritional counseling) vs. Metformin vs. placebo	Lifestyle intervention group achieved a 38% reversal of MetSyn and a 41% reduction of new onset MetSyn.

PA—physical activity; LTPA—leisure time physical activity; MetSyn—metabolic syndrome; HDL—high density lipoprotein; LDL—high density lipoprotein; CVD—cardiovascular disease; HbA1c—glycated hemoglobin.

**Table 2 nutrients-11-01652-t002:** Sampling of studies assessing the association between cardiorespiratory fitness and the metabolic syndrome.

Author, Year; (Reference)	*N* (Men/Women)	Key Results
Carnethon, 2003; [49]	4487 (2029/2458)	Only men and women in the highest 40% of maximal treadmill performance were protected against developing MetSyn.
Franks, 2004; [50]	847 men	A strong inverse association between physical activity and MetSyn. The magnitude of the association between physical activity and the MetSyn was >3-fold greater than for VO_2_max.
LaMonte, 2005; [46]	10,498 (9007/1491)	An independent and progressive decline in the risk of developing MetSyn with higher CRF for men and women. Also, 20% to 26% lower risks occurred among participants with moderate CRF and 53% to 63% lower risks observed in highest CRF categories vs. the lowest CRF category.
Hassinen, 2008; [44]	1347 (671/676)	Men and women in the lowest third of VO_2_max had 10.2 times (men) and 10.8 times (women) higher risk of having MetSyn than those in the highest VO_2_max category.
Hassinen, 2010; [48]	1226 (589/637)	Risk of developing MetSyn within 2 years of follow-up was 44% lower for each 1-SD increase in VO_2_ max. Each 1-SD higher VO_2_ max from baseline resulted in 1.8 times higher likelihood to resolve MetSyn during 2 years of follow-up.
Earnest, 2013; [51]	38,659 (30,927/7732)	CRF demonstrated a strong inverse relationship with MetSyn in both genders. The association was strongest in those with lower waist circumference and fasting glucose, in both genders.
Adams-Campbell, 2016; [47]	170 women	CRF was inversely related to the prevalence of the metabolic syndrome in overweight/obese African-American postmenopausal women.
Ingle, 2017; [52]	9666 men	The likelihood of developing MetSyn was approximately 50% lower in fit men compared to unfit, independent of BMI particularly in men <50 years.
Kelly, 2018; [45]	3636 (2007/1629)	Significant, inverse and graded association between VO_2_max and MetSyn. Highest fit had >20 times lower risk of having MetSyn compared to least-fit individuals. The difference in VO_2_max between those with MetSyn and those without was ≈ 2.5 METs.

CRF—cardiorespiratory fitness; BMI—body mass index; MetSyn—metabolic syndrome; METS—metabolic equivalents.

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
