# Peer review of "Physical Activity, Cardiorespiratory Fitness, and the Metabolic Syndrome"

_nutrients, 2019, doi:10.3390/nu11071652_

Round 1
Reviewer 1 Report
The institutional affiliations are not complete, they must indicate the institution and the departments where the researchers are housed.
General recommendation of the article: the article must be synthesized, and some concepts clarified. The excess of information makes the article is confused as it is in this case, making an interesting synthesis and making the appropriate changes the article could be of great interest.
Abstract: from my point of view it is repetitive, it must be specific and direct in the message that it is wanted to transmit, because in several occasions it repeats that the physical activity. In addition, it is not a good message to convey that an intervention alone can solve a pathology or a syndrome like this.
Development of the article:
I recommend updating the WHO approach, for a more current reference (for example: The World Health Organization STEPwise Approach to Noncommunicable Disease Risk-Factor Surveillance: Methods, Challenges, and Opportunities, Am J Public Health, 2016; 106 (1): 74-8, doi: 10.2105 / AJPH.2015.302962). This reference clearly indicates the importance of avoiding physical inactivity in chronic noncommunicable diseases.
Reference number 2 should be updated, by a more current one (for example: Pucci G, Sex- and gender-related prevalence, cardiovascular risk and therapeutic approach in metabolic syndrome: A review of the literature, Pharmacol Res. 2017; 120: 34 -42.doi: 10.1016 / j.phrs.2017.03.008).
In the first paragraph of page 1 you talk about components of the metabolic syndrome, it is not an adequate term, you must change it by diagnostic criteria, in addition to that you must indicate a bibliographic citation where you indicate the different diagnostic criteria, avoiding adhering a table in this publication.
In the first paragraph of the first page, you use the term cardiorespiratory fitness, you must define it. As the definition of metabolic syndrome should be at the beginning of the article, along with the indication of the diagnostic criteria discussed. I recommend introducing the term of physical activity, physical exercise, sedentary subject and active subject so that researchers who do not have previous knowledge know the depth of this topic.
In the second paragraph of the first page, it is not clear what role cardiorespiratory fitness plays in decreasing the prevalence of the diagnostic criteria of the metabolic syndrome. The indicated variations are caused by the realization of physical activity.
In paragraph of the second page, you indicate the cardiometabolic term, any included term must be defined.
Last paragraph of the second page, mix the concept of physical activity and physical exercise as if it were the same, I recommend that they define the terms and apply according to their definition.
Table 1 must be presented in another way, it must be indicated how the search was performed and the patterns of the search. I provide an article so that you are how: New insights about how to make an intervention in children and adolescents with metabolic syndrome: diet, exercise vs changes in body composition. A systematic review of RCT. Nutrients 2018; 10 (878): 1-24. DOI: 10.3390 / nu10070878. The information of interest is in section 2.1. Indicate why those types of articles are selected (observational) and not others.
The section "Physical Activity and the Metabolic Syndrome" must be changed according to the previous recommendations. In addition, it is recommended to make a synthesis of the information obtained, focused on the changes produced according to what type they are.
The changes recommended above should also be extended to the information presented in Table 2.
In the indications of the information in Table 1 and 2, it should be clarified in a very specific way what level of exercise intensity is what helps to improve the diagnostic criteria and to what extent it changes them. The levels of intensity should be described by objective and specific data as indicated by the article of American College of Sports Medicine (American College of Sports Medicine position stand). apparently healthy adults: guidance for prescribing exercise Med Sci Sports Exerc. 2011; 43 (7): 1334-59. doi: 10.1249 / MSS.0b013e318213fefb.).
The section that explains the relationship between the Cardiorespiratory Fitness and the Metabolic Syndrome, should be synthesized and made more specific so that this relationship can be observed. However, it does not show why this relationship should exist and not exercise with intermittency or others.
The section of “Mechanisms Underlying the Metabolic Syndrome and Implications for Physical Activity and Fitness”, should be synthesized and also, I would put it at the beginning of the article, not at the end. In addition, this section is the Mechanisms Underlying the Metabolic Syndrome, but not its relation to physical activity.
Regarding the summary of this article, it should be improved based on the improvement of the previous sections.
Bibliography:
1. There are different types of letters and sizes.
2. Identified some cases without indicating the year of said reference.
3. The style of how to indicate the bibliography is not followed, I recommend that you review the instructions of the authors and review an article of this recently published journal.
4. There are some quotes that I do not even want to have the original format of Vancouver apicado correctly.
5. 58% of the bibliographic references are older than 5 years, in addition, there are many citations with more seniority of 10 years. This is a quarter 1 magazine in the area of nutrition and dietetics, the bibliography being a vital engine to produce a great design of a review or other article.
Author Response
Comment. The institutional affiliations are not complete, they must indicate the institution and the departments where the researchers are housed.
Response. As the reviewer suggests, the specific department affiliation for each of the authors is now included.
Comment. General recommendation of the article: the article must be synthesized, and some concepts clarified. The excess of information makes the article is confused as it is in this case, making an interesting synthesis and making the appropriate changes the article could be of great interest.
Response. The English in this comment is a bit unclear to us, but we have added a synthesis of the information following each section as it appears that the reviewer is requesting.
Comment. Abstract: from my point of view it is repetitive, it must be specific and direct in the message that it is wanted to transmit, because in several occasions it repeats that the physical activity. In addition, it is not a good message to convey that an intervention alone can solve a pathology or a syndrome like this.
Response. We believe the reviewer has mis-read the abstract. We explicitly stated that one intervention alone cannot resolve the metabolic syndrome. See lines 7-11: “While physical activity interventions that lead to improved fitness cannot be expected to normalize insulin resistance, lipid disorders, or obesity, the combined effect of increasing activity on these risk markers, an improvement in fitness, or both, has been shown to have a major impact on health outcomes related to the metabolic syndrome”. We don’t see any repetition in the abstract, and the overall message appears to be explicit and clear.
Comment. I recommend updating the WHO approach, for a more current reference (for example: The World Health Organization STEPwise Approach to Noncommunicable Disease Risk-Factor Surveillance: Methods, Challenges, and Opportunities, Am J Public Health, 2016; 106 (1): 74-8, doi: 10.2105 / AJPH.2015.302962). This reference clearly indicates the importance of avoiding physical inactivity in chronic noncommunicable diseases.
Reference number 2 should be updated, by a more current one (for example: Pucci G, Sex- and gender-related prevalence, cardiovascular risk and therapeutic approach in metabolic syndrome: A review of the literature, Pharmacol Res. 2017; 120: 34 -42.doi: 10.1016 / j.phrs.2017.03.008).
Response. We agree with the reviewer regarding the WHO reference, and have added this reference as suggested. Please note that reference #2 refers specifically to Dr. Haller’s seminal paper; he is credited by some as the first to formally use the term “metabolic syndrome”. The Pucci reference suggested by the reviewer is an excellent one, and has been added in the following sentence.
Comment. In the first paragraph of page 1 you talk about components of the metabolic syndrome, it is not an adequate term, you must change it by diagnostic criteria, in addition to that you must indicate a bibliographic citation where you indicate the different diagnostic criteria, avoiding adhering a table in this publication.
Response. The definitions of metabolic syndrome have a well-recognized lengthy and controversial history. As mentioned above, given the context of the larger symposium and the fact that these issues will be covered elsewhere, we felt that it was not appropriate to address the many diagnostic criteria in our article given the limited space.
Comment. In the first paragraph of the first page, you use the term cardiorespiratory fitness, you must define it. As the definition of metabolic syndrome should be at the beginning of the article, along with the indication of the diagnostic criteria discussed. I recommend introducing the term of physical activity, physical exercise, sedentary subject and active subject so that researchers who do not have previous knowledge know the depth of this topic.
Response. As suggested, we have added a definition of cardiorespiratory fitness in the first paragraph. The reviewer is correct in that physical activity is defined in many different ways. As we discussed each article throughout our paper, we used the definition that each respective study used. Physical activity may have been defined by quartiles or quintiles, at times dichotomous (eg. meeting vs. not meeting the minimal guidelines for physical activity), or by overall energy expended. As the reviewer suggests, we have added definitions of “physical activity” and “exercise” in the first paragraph of the introduction.
Comment. In the second paragraph of the first page, it is not clear what role cardiorespiratory fitness plays in decreasing the prevalence of the diagnostic criteria of the metabolic syndrome. The indicated variations are caused by the realization of physical activity.
Response. We are uncertain as to what the reviewer is referring to here; we will be happy to revise accordingly if this is clarified for us.
Comment. In paragraph of the second page, you indicate the cardiometabolic term, any included term must be defined. Last paragraph of the second page, mix the concept of physical activity and physical exercise as if it were the same, I recommend that they define the terms and apply according to their definition.
Response. As the reviewer suggests, we have added brief definitions and a distinction between physical activity and exercise to first paragraph in the introduction. Again, it seems to us that defining the specifics of the metabolic syndrome (which are historically complex and highly nuanced) is not a good use of space given that this issue is dealt with in other papers in the symposium.
Comment. Table 1 must be presented in another way, it must be indicated how the search was performed and the patterns of the search. I provide an article so that you are how: New insights about how to make an intervention in children and adolescents with metabolic syndrome: diet, exercise vs changes in body composition. A systematic review of RCT. Nutrients 2018; 10 (878): 1-24. DOI: 10.3390 / nu10070878. The information of interest is in section 2.1. Indicate why those types of articles are selected (observational) and not others.
The section "Physical Activity and the Metabolic Syndrome" must be changed according to the previous recommendations. In addition, it is recommended to make a synthesis of the information obtained, focused on the changes produced according to what type they are.
The changes recommended above should also be extended to the information presented in Table 2.
Response. As mentioned above, our purpose was not by any means to conduct a systematic review or an exhaustive “search”. The paper cited by the reviewer is a systematic review that includes 11 pages of tables and a number of figures. Rather, we endeavored only to provide an overview of this issue using a handful of key studies in order to give the reader a sense of the impact of physical activity on the metabolic syndrome. A systematic review would require a prohibitively lengthy article and numerous pages of tables.
Comment. In the indications of the information in Table 1 and 2, it should be clarified in a very specific way what level of exercise intensity is what helps to improve the diagnostic criteria and to what extent it changes them. The levels of intensity should be described by objective and specific data as indicated by the article of American College of Sports Medicine (American College of Sports Medicine position stand). apparently healthy adults: guidance for prescribing exercise Med Sci Sports Exerc. 2011; 43 (7): 1334-59. doi: 10.1249 / MSS.0b013e318213fefb.).
Response. The available articles on physical activity, fitness, and their relationship to the metabolic syndrome were not designed in a way that specified which particular exercise intensity improved a particular diagnostic criterion for the metabolic syndrome. The majority of the exercise programs followed standard guidelines on exercise prescription. Again, some of these details extend beyond the scope of our paper; specifics on the different exercise programs are presented in the text and tables as they are provided by the authors of the various papers.
Comment. The section that explains the relationship between the Cardiorespiratory Fitness and the Metabolic Syndrome, should be synthesized and made more specific so that this relationship can be observed. However, it does not show why this relationship should exist and not exercise with intermittency or others.
Response. As the reviewer suggests and as mentioned above, we have added some text at the end of each of the fitness and physical activity sections that synthesizes the studies. By “intermittency”, we are uncertain if the reviewer is requesting a review of interval training. If so, the literature base is extremely limited in this area.
Comment. The section of “Mechanisms Underlying the Metabolic Syndrome and Implications for Physical Activity and Fitness”, should be synthesized and also, I would put it at the beginning of the article, not at the end. In addition, this section is the Mechanisms Underlying the Metabolic Syndrome, but not its relation to physical activity.
Response. As suggested by the reviewer, we have added a synthesis of the studies following each section. The co-authors have discussed moving the section on mechanisms to the beginning, but we felt that since the paper focuses on physical activity and fitness, it would be preferable to keep the order of the sections the way they are. We can rearrange the sections if the senior editors feel strongly about this.
Comment; Bibliography:
1. There are different types of letters and sizes.
2. Identified some cases without indicating the year of said reference.
3. The style of how to indicate the bibliography is not followed, I recommend that you review the instructions of the authors and review an article of this recently published journal.
4. There are some quotes that I do not even want to have the original format of Vancouver apicado correctly.
Response. The suggested changes to the bibliography have been made in the revision; all references now meet the specifications of the journal.
Reviewer 2 Report
This review of physical activity, cardiorespiratory fitness and the metabolic syndrome seeks to summarise the current evidence that physical activity and higher fitness mitigates metabolic syndrome.
The review is not stated to be systematic and does state that an overview of effects is provided along with discussion of the possible mechanisms. The discussion of mechanisms is limited as is the measures of CRF – not just VO2 max or peak. I would have liked to see more about muscle physiology.
It is a narrative review of observation, intervention and meta analyses but some of the material presented is repeated and there is lack of critique. However it would be helpful to see how the authors undertook their literature search to provide this overview as many of the references are dated..
There are no line numbers on the pdf – so I use page numbers and paragraphs.
Throughout when talking about impact or effect please state the direction of the impact – favourable or not and describe the magnitude
Table 1 Please refer to participants with diabetes not as diabetics – people first language https://en.wikipedia.org/wiki/People-first_language For the first study PA is stated to be by self report but the next two the determination of PA is not described.
Table 1 the dose of physical activity would be useful to compare interventions i.e. time, frequency and intensity and calendar time (which is reported) if available. The age and gender of participants would also add critique to this table
Page 3 paragraph 2 onwards Thune and 2 other studies are described in more detail after summary in table 1 – there is therefore repeat of information in the table and in the text.
References in Table 1 do not have number so cannot match to reference list easily – Laaksonen 2002 with 612 men over 7 years in table but reported as follow up at 4 years in the text
Words like dramatic, improved and worsened to not give a measure of the effect size or the clinical effect – rather than statistical significance.
Only one study of PA used accelerometry which is an objective measure
What was the difference in BMI and lipids in the Ekelund study?
Suggest either expanding Table 1 to include more detail or moving to supplementary material or deleting and combining with Table 2 . There is repetition of studies in Table 1 in table 2. Table 1 is measures of physical activity and table 2 cardiorespiratory fitness. The Laaksonen study is stated as 2001 in Table 2 and 2002 in the text. How metabolic syndrome was defined e.g. WHO ATP IDF should be stated for each study as well as age of participants.
Table 2 it is not clear if it is cross sectional studies or cohort - Laaksonen was 4 year follow-up.
Page 4 Statements like “Vigorous LTPA had an even stronger inverse association among men who were unfit at baseline.” Need to be more explicit – inverse association with what? Presumably metabolic syndrome but how defined – 4 factors or 1?
Page 7 Self reported PA and measured physical fitness (CRF) are discussed but the strength of association is not clear
Check the manuscript throughout for plagiarism
Katzmarzyk,et al (34) studied the efficacy of exercise training in treating the metabolic syndrome among 621 participants from the HERITAGE Family Study, identified at baseline as sedentary but apparently healthy. The presence of the metabolic syndrome and component risk factors were determined before and after 20 weeks of supervised aerobic exercise training. Of the 105 participants with the metabolic syndrome at baseline, 30.5% (32 participants) were no longer classified as having the metabolic syndrome after training. Among the 32 participants who improved their metabolic profile, 43% decreased triglycerides, 16% improved HDL cholesterol, 38% decreased blood pressure, 9% improved fasting plasma glucose, and 28% decreased their waist circumference. There were no sex or race differences in the efficacy of exercise in treating the metabolic syndrome: 32.7% of men, 28.0% of women, 29.7% of black, and 30.9% of white participants with the metabolic syndrome were no longer classified as having the syndrome after training.
Is very similar, copied in places, to the abstract of Katzmarzyks paper.
Page 8 Hassinen the value for women was 4.8ml/kg/min higher VO2 max for women to reduce risk 35% - the value for mean has been swapped. These women and men were aged 57-78 years which is important when discussing gender differences – as most women would have experienced menopause
The section on mechanisms does not discuss muscle physiology in any depth. Physical fitness is adaptation of skeletal muscle to be metabolically efficient e.g. increase in number of mitochondria, myoglobin, vascularisation all improve metabolism and reduce insulin resistance i.e. upregulate Glut 4.. Muscle fitness reduces the need for insulin – taking the load off the pancreas.
The importance of diet in metabolic syndrome is not mentioned at all
Minor – examples.
Page 2 para 2 the sentence that includes “unfortunate given the strength of exercise interventions on health outcomes” does not scan – the strength of ?effect and ?beneficial health outcomes.
Table 1 Assessment not “assessment”
Page 4 paragraph 1 Increasing has a capital I but not start of sentence.
Table 2 – three studies are mainly or entirely men – how do the findings for men and women compare?
DM2 expand abbreviations first time used and be consistent throughout.
Author Response
This review of physical activity, cardiorespiratory fitness and the metabolic syndrome seeks to summarise the current evidence that physical activity and higher fitness mitigates metabolic syndrome. The review is not stated to be systematic and does state that an overview of effects is provided along with discussion of the possible mechanisms. The discussion of mechanisms is limited as is the measures of CRF – not just VO2 max or peak. I would have liked to see more about muscle physiology.
Comment. It is a narrative review of observation, intervention and meta analyses but some of the material presented is repeated and there is lack of critique. However it would be helpful to see how the authors undertook their literature search to provide this overview as many of the references are dated.
Response. As mentioned above, we did not attempt to conduct a systematic review or meta-analysis, and the article was not intended to be comprehensive. Such an article would be prohibitively lengthy. As mentioned in response to reviewer #1, we have added a summary/critique following each section as suggested by the reviewer.
Comment. Throughout when talking about impact or effect please state the direction of the impact – favourable or not and describe the magnitude
Response. As the reviewer suggests, we have reviewed the manuscript and more explicitly stated the direction and magnitude of the impact of physical activity and fitness on the metabolic syndrome.
Comment. Table 1 - Please refer to participants with diabetes not as diabetics – people first language https://en.wikipedia.org/wiki/People-first_language. For the first study PA is stated to be by self report but the next two the determination of PA is not described.
Response. “Diabetics” has been changed to participants with diabetes as suggested. The second and third studies in Table 1 are self-reported PA and this has been clarified in the revision.
Comment. Table 1 the dose of physical activity would be useful to compare interventions i.e. time, frequency and intensity and calendar time (which is reported) if available. The age and gender of participants would also add critique to this table.
Response. We agree that adding the proportion of men and women to Table 1 would be informative; this has been added in the revision. As we have presented in the text, the time, frequency, and intensity of the exercise interventions do not differ appreciably and are most often designed to meet or exceed the physical activity guidelines. Mean age of the participants has also been added to the table as suggested.
Comment. Page 3 paragraph 2 onwards Thune and 2 other studies are described in more detail after summary in table 1 – there is therefore repeat of information in the table and in the text.
Response. There are many additional studies that could be added to both Table 1 and the text – it would not be feasible to include all of them in detail. As mentioned above, our objective was to provide an overview of the issue by choosing some key studies in order to give the reader a sense of the impact of physical activity interventions on the metabolic syndrome.
Comment. References in Table 1 do not have number so cannot match to reference list easily – Laaksonen 2002 with 612 men over 7 years in table but reported as follow up at 4 years in the text.
Response. Reference numbers and year of publication have been added to Table 1 in the revision. The Laaksonen study was a 4-year-follow-up, and this has been clarified in the text.
Comment. Words like dramatic, improved and worsened to not give a measure of the effect size or the clinical effect – rather than statistical significance.
Response. As mentioned above, given the volume of studies in this area, our purpose was to be more conversational rather than rigid and comprehensive. Nevertheless, in the revision we have lessened the use of adjectives when describing the studies.
Comment. What was the difference in BMI and lipids in the Ekelund study?
Response. The Ekelund study presented their data as associations with different levels of physical activity; the precise differences in risk markers were not presented.
Comment. Suggest either expanding Table 1 to include more detail or moving to supplementary material or deleting and combining with Table 2. There is repetition of studies in Table 1 in table 2. Table 1 is measures of physical activity and table 2 cardiorespiratory fitness. The Laaksonen study is stated as 2001 in Table 2 and 2002 in the text. How metabolic syndrome was defined e.g. WHO ATP IDF should be stated for each study as well as age of participants.
Response. Because physical activity (a behavior) and fitness (an attribute) are two distinct entities, we felt it would be confusing to put both types of studies into one table. While the Laaksonen study included both physical activity and fitness, it has been removed from Table 2 in the revision; it is more appropriate to include it in the list of physical activity studies in Table 1. No other studies are repeated in the two tables. As mentioned above, given the complexity and the many definitions of the metabolic syndrome, we felt it was not appropriate to address this in our paper. Mean age for each study has been added in both Tables 1 and 2 as suggested.
Comment. Table 2 it is not clear if it is cross sectional studies or cohort - Laaksonen was 4 year follow-up.
Response. The Laaksonen study is in Table 1 only in the revision, and we have clarified the fact that the Laaksonen study was longitudinal.
Comment. Page 4 Statements like “Vigorous LTPA had an even stronger inverse association among men who were unfit at baseline.” Need to be more explicit – inverse association with what? Presumably metabolic syndrome but how defined – 4 factors or 1?
Response. We have clarified this statement on page 4 and elsewhere so that these associations are more explicit. As mentioned above, we would prefer not to get into specifics regarding how the metabolic syndrome was defined since this issue is complex and is discussed in other papers in the symposium.
Comment. Page 7 Self-reported PA and measured physical fitness (CRF) are discussed but the strength of association is not clear.
Response. We are uncertain as to which study the reviewer is referring to as we don’t believe that page 7 is correct. However, as mentioned above, we have attempted to more explicitly state the strength of the associations between PA, CRF and metabolic risk in the revision.
Comment. Check the manuscript throughout for plagiarism
Katzmarzyk,et al (34) studied the efficacy of exercise training in treating the metabolic syndrome among 621 participants from the HERITAGE Family Study, identified at baseline as sedentary but apparently healthy. The presence of the metabolic syndrome and component risk factors were determined before and after 20 weeks of supervised aerobic exercise training. Of the 105 participants with the metabolic syndrome at baseline, 30.5% (32 participants) were no longer classified as having the metabolic syndrome after training. Among the 32 participants who improved their metabolic profile, 43% decreased triglycerides, 16% improved HDL cholesterol, 38% decreased blood pressure, 9% improved fasting plasma glucose, and 28% decreased their waist circumference. There were no sex or race differences in the efficacy of exercise in treating the metabolic syndrome: 32.7% of men, 28.0% of women, 29.7% of black, and 30.9% of white participants with the metabolic syndrome were no longer classified as having the syndrome after training.
Is very similar, copied in places, to the abstract of Katzmarzyks paper.
Response. The description of the Katzmarzyk paper has been modified such that it is less similar to the abstract.
Comment. Page 8 Hassinen the value for women was 4.8ml/kg/min higher VO2 max for women to reduce risk 35% - the value for mean has been swapped. These women and men were aged 57-78 years which is important when discussing gender differences – as most women would have experienced menopause.
Response. The reviewer is correct, and we have clarified this in the revision; it now more accurately reads: “Hassinen et al (44) observed that each 1-SD higher VO2 max (6.1 ml/kg/min in men; 4.8 ml/kg/min in women) resulted in 56% and 35% decreased risks of developing metabolic syndrome over two years of follow-up in men and women, respectively”.
Comment. The importance of diet in metabolic syndrome is not mentioned at all.
Response. There are 6 articles in this special issue that focus on diet. Since the focus of our paper was on fitness and physical activity, we felt it would be out of context to cover issues related to diet.
Comments - Minor
Page 2 para 2 the sentence that includes “unfortunate given the strength of exercise interventions on health outcomes” does not scan – the strength of ?effect and ?beneficial health outcomes.
Table 1 Assessment not “assessment”
Page 4 paragraph 1 Increasing has a capital I but not start of sentence.
DM2 expand abbreviations first time used and be consistent throughout.
Response. The text has been modified in each of these instances to improve clarity.
Comment. Table 2 – three studies are mainly or entirely men – how do the findings for men and women compare?
Response. As mentioned above, the purpose of Table 2 was to provide an overview of some of the representative studies on fitness and its impact on the metabolic syndrome. While the effect appears to be similar between men and women, studies specifically comparing men and women are lacking. As mentioned above, we have specified the proportion of men and women in the studies in Tables 1 and 2.
Reviewer 3 Report
n/a;Excellent overview is provided, along with a discussion of the mechanisms underlying the benefits of being more fit or more physically active in the prevention and treatment of the metabolic syndrome.
The review is very interesting and timely, with thorough discussion of advantages and weaknesses of the current published studies reviewed.
Author Response
Comments:
Excellent overview is provided, along with a discussion of the mechanisms underlying the benefits of being more fit or more physically active in the prevention and treatment of the metabolic syndrome.
The review is very interesting and timely, with thorough discussion of advantages and weaknesses of the current published studies reviewed.
Response.:
Thanks.
Round 2
Reviewer 1 Report
The recommendations indicated in the previous review round were part of an in-depth review, based on the fact that the article obtained important improvements in the bibliography and a summary of the presented information was made. In my opinion, these recommendations for change have not been made in full.
The instructions in this journal are clear, accurate and concise information must be provided, along with using the PRISMA system guide.
Author Response
To the best of our knowledge, the journal guidelines have been followed. Please let us know if there are any specific changes needed in terms of these guidelines or formatting of the article.
Reviewer 2 Report
Thank you for the responses. Not all changes were in red text so it was difficult to check all responses but in particular the statement that "The description of the Katzmarzyk paper has been modified such that it is less similar to the abstract" is not true. There are no changes - was an earlier version of the revised manuscript uploaded.
Author Response
The reviewer has requested that we change the description of the Katzmarzyk paper so that it is less similar to the abstract. In the revision, we have re-written this paragraph so that this is the case.